# Perceived Health Status Predicts Resilience after Hip Fracture in Older People

**DOI:** 10.3390/medicina60101621

**Published:** 2024-10-03

**Authors:** Diana Lelli, Maria Serena Iuorio, Raffaele Antonelli Incalzi, Claudio Pedone

**Affiliations:** 1Operative Research Unit of Geriatrics, Fondazione Policlinico Universitario Campus Bio-Medico, 00128 Rome, Italy; d.lelli@policlinicocampus.it; 2Research Unit of Geriatrics, Università Campus Bio-Medico di Roma, 00128 Rome, Italy; mariaserena.iuorio@unicampus.it; 3Research Unit of Internal Medicine, Università Campus Bio-Medico di Roma, 00128 Rome, Italy; r.antonelli@policlinicocampus.it; 4Operative Research Unit of Internal Medicine, Fondazione Policlinico Universitario Campus Bio-Medico, 00128 Rome, Italy

**Keywords:** resilience, hip fracture, rehabilitation, older, perceived health status

## Abstract

*Background and Objectives:* Perceived health status (PHS) is associated with various health outcomes in older adults, but its relationship with resilience in the context of events with a major impact on functional status (FS), such as hip fracture, has not been explored. Our objective was to evaluate whether older adults who report good PHS before a hip fracture have a higher probability of returning to their baseline physical performance (PP) and personal independence. *Materials and Methods:* We analyzed data from waves 1 and 2 of the Survey of Health, Ageing and Retirement in Europe (SHARE) study, enrolling patients ≥ 65 years who experienced a hip fracture between these two waves. As study outcomes, we analyzed changes in PP and functional abilities (FAs). *Results:* We included 149 participants with a mean age of 75.7 years (SD: 6.5); women comprised 66%. The incidence of loss of PP was 51.7% among participants with good PHS and 59.6% among those with poor PHS. FA worsened in 40% of participants with good PHS and 58.4% in those with poor PHS. Relative risk (RR) for loss of FA in people with good PHS was 0.68 (95% CI: 0.48–0.98), which did not change after an adjustment for age, gender, baseline FA, depression, number of comorbidities, education, income, and social support, despite it not reaching statistical significance. After adjustment, the risk of worsening PP in participants with good PHS was reduced by 34% (95% CI: 0.41–1.06). *Conclusions:* A simple question on PHS may predict the resilience of older adults after an acute stressor. A systematic evaluation of PHS can help identify patients with a higher probability of regaining function after a hip fracture and thus provide useful information for resource allocation.

## 1. Introduction

Aging is associated with an increased susceptibility to events that can significantly impact physical and mental well-being. A key factor to maintain health and function across the lifespan is to minimize the negative effects of injury, illness, and other stressors that inevitably occur in life. Resilience has been increasingly recognized as a crucial determinant in how older adults cope with these challenges [1,2]. Resilience is a multidimensional and dynamic process that reflects the ability to adapt to adversity and recover or optimize function in the face of age-related losses or disease [3]. This ability is shaped by the complex interplay between physical, psychological, and environmental factors that influence recovery trajectories after stressful events [4,5,6].

Perceived health status (PHS) is a subjective assessment that reflects an individual’s overall sense of health and well-being [7]. It has been shown that PHS is linked with health outcomes in older adults, including mortality [8,9] and risk of hospitalization [10]. We have also previously shown that PHS can be used to predict resilience in older people after a self-reported stressful health event [11].

To the best of our knowledge, however, a possible relationship between health self-perception and resilience in the context of events with a major impact on functional status, such as hip fracture, has not been explored. Hip fracture is a major concern in older people, with a global annual prevalence increasing with the aging of the population and expected to reach 4.5 million by 2050 [12]. These fractures have a serious impact on morbidity and mortality, with around 5% of patients dying within a month and 19% within a year. It is also estimated that up to half of the patients fail to regain their pre-fracture levels of mobility and independence [13,14]. Hence, understanding the factors that can influence recovery outcomes in these patients is essential.

Our hypothesis is that people with a good PHS may be more likely to regain their previous functional status after a hip fracture. We assume that individuals who perceive their health positively are more likely to exhibit greater resilience, hence maintaining higher levels of independence post fracture. Understanding this relationship may help to identify factors that can be supported to enhance resilience in older adults and develop targeted strategies to enhance recovery and quality of life for older patients experiencing hip fractures.

Therefore, the objective of this study is to evaluate whether older adults who report good PHS before a hip fracture have a higher probability of returning to their baseline physical performance and personal independence compared to those who report fair or poor health status.

## 2. Methods

### 2.1. Data Source

We used data from the Survey of Health, Ageing and Retirement in Europe (SHARE) study [15], a multidisciplinary and cross-national panel database of micro data on health, socio-economic status and social and family networks of more than 140,000 individuals from 28 European countries (+Israel) aged 50 or over. The survey is extensively explained elsewhere [16]. Briefly, the study started in 2004 and data collection was conducted biennially through face-to-face interviews using a rigorous multistage clustered sampling method. Exclusion criteria were death before the starting of the field period, incarceration, hospitalization or being out of the country during the entire survey period, an inability to speak the country’s language(s), or to have moved to an unknown address. A proxy respondent, usually a family member, a household member, a neighbor, or another person related to the social network, was allowed to answer if the respondent was unable to complete the interview due to physical or mental health limitations [17].

For the purposes of this study, we used data from SHARE wave 1 (2004–2006) and wave 2 (2006–2007) (DOIs: 10.6103/SHARE.w1.900, 10.6103/SHARE.w2.900). The SHARE study is subject to a continuous ethics review. During waves 1–4, SHARE was reviewed and approved by the Ethics Committee of the University of Mannheim. Wave 4 and the continuation of the project were reviewed and approved by the Ethics Council of the Max Planck Society.

### 2.2. Sample Selection

We included participants aged 65 and over who experienced a hip fracture between the first and second waves of the study and for whom baseline PHS data were available. Hip fracture status was assessed at wave 2 by asking the question “Have you suffered a hip fracture since we last interviewed you?”.

### 2.3. Measure of Exposure

PHS was assessed by asking the question “Would you say your health is” and using two randomly assigned Likert scales for the answer: Excellent/Very Good/Good/Fair/Poor and Very Good/Good/Fair/Bad/Very Bad.

To compare people with good vs. poor PHS, both scales were dichotomized using “Excellent”, “Very Good”, and “Good” rates as indicators of good PHS and “Fair”, “Poor”, “Bad”, and “Very Bad” as indicators of poor PHS. By dichotomizing this variable, we were able to compare participants with good vs. those with poor PHS.

### 2.4. Study Outcomes

Our primary outcome was the change in functional status between waves 1 and 2. Functional status was self-reported and explored with two different questionnaires. The first one explored physical performance by asking the following question: “Because of a health problem, do you have difficulty doing any of the activities on this card? Exclude any difficulties that you expect to last less than three months.” (1. walking 100 m, 2. sitting for about two hours, 3. getting up from a chair after sitting for long periods, 4. climbing several flights of stairs without resting, 5. climbing one flight of stairs without resting, 6. stooping, kneeling, or crouching, 7. reaching or extending your arms above shoulder level, 8. pulling or pushing large objects like a living room chair, 9. lifting or carrying weights over, 10. lifting pounds/5 kg, like a heavy bag of groceries, and 11. picking up a small coin from a table). For each task, participants indicated whether or not they had difficulty performing the task (yes = 1; no = 0). The score ranges from 0 to 11, with higher scores indicating a lower degree of physical performance (“physical performance score”).

The second questionnaire explored subjects’ ability to perform activities of daily living by asking the following question: “Because of a health or memory problem, do you have difficulty doing any of the activities on this card? Exclude any difficulties that you expect to last less than three months.” (1. dressing, including putting on shoes and socks, 2. walking across a room, 3. bathing or showering, 4. eating, such as cutting up your food, 5. getting in or out of bed, 6. using the toilet, including getting up or down, 7. using a map to figure out how to get around in a strange place, 8. preparing a hot meal, 9. shopping for groceries, 10. making telephone calls, 11. taking medications, 12. carrying out work around the house or garden, and 13. managing money, such as paying bills and keeping track of expenses). For each question, participants indicated whether they had difficulty performing the task (yes = 1; no = 0). The overall score ranges from 0 to 13, with a higher score indicating a higher degree of disability (“functional ability score”).

We calculated the difference in these scores between the two interviews. We also dichotomized the scores using a cut-off value of 1, indicating the loss during the follow-up of the capacity for performing at least one task related to physical performance or functional ability.

### 2.5. Study Variables

The socio-demographic variables considered were age, sex, annual net income, marital status and education, measured according to the International Standard Classification of Educational Degrees (ISCED-97); participants were categorized as having or not having upper secondary education (ISCED code 3 or above).

Depression was assessed through the EURO-D scale [18,19], a validated scale that includes 12 items: depressed mood, pessimism, wishing for death, guilt, sleep, interest, irritability, appetite, fatigue, concentration, enjoyment, and tearfulness. Each item is coded as 0 (symptom absent) and 1 (symptom present), with a total score ranging from 0 to 12, with higher values indicating more depressive symptoms present in the respondent.

The number of coexisting chronic diseases was self-reported and assessed through a question about the presence of cardiovascular diseases (including myocardial infarction, coronary thrombosis, congestive heart failure, high blood pressure or hypertension, and high blood cholesterol), cerebral vascular disease, diabetes or high blood sugar, chronic lung disease (including chronic bronchitis, emphysema, and asthma), osteoarticular diseases, cancer or malignant tumor (including leukemia or lymphoma), stomach or duodenal ulcer, Parkinson’s disease, cataracts, and hip fracture.

We also analyzed drug therapy using a question about therapy prescribed for 15 chronic conditions (e.g., high blood pressure, diabetes, etc.).

The presence of social support was assessed through the question “Thinking about the activities that you have problems with, does anyone (including cohabiting caregivers, family members from outside the household and any friend or neighbor) ever help you with these activities?”.

### 2.6. Analytic Approach

People with good vs. poor PHS were compared with respect to demographic, gender, education, the number of comorbidities, the number of diseases pharmacologically treated, social support, physical performance, and functional ability using descriptive statistics (means and standard deviations for continuous variables and proportions for categorical variables).

The changes over time in physical performance and functional ability in people with good vs. poor PHS were compared using the T-test for independent samples. The association between good PHS and the loss of physical performance ability and of functional ability was evaluated using relative risks with 95% CI; these estimates were adjusted for potential confounders (age, gender, financial income, number of comorbidities, depression, and social support) and a baseline score using a Poisson regression with a robust confidence interval estimation.

All the analyses were performed using R Studio version 4.2.0 for Windows (R Foundation for Statistical Computing, Vienna, Austria).

## 3. Results

In the second wave of the SHARE study, from the 37,132 participants that responded to the survey, we identified 149 participants with an age ≥ 65 years who had suffered a hip fracture between the first and the second wave and for which information on the PHS at baseline was available (Figure 1).

The mean age of the study population was 75.7 years (SD: 6.5), and women comprised 66%. The mean follow-up time was 30.2 months (SD: 7.5). Patients with poor PHS (*n* = 89) were more frequently female (68.5% vs. 63.3%), had a higher number of diseases (2.18, SD 1.3 vs. 3.13, SD 1.8), and received more pharmacological treatments (1.62, SD 1.2 vs. 3.31, SD 2.0). They also had a worse functional ability score (2.7, SD 3.5 vs. 0.7, SD 1.8) and a worse physical performance score (4.3, SD 2.7 vs. 2.1, SD 1.9) (Table 1).

The distribution of the physical performance and functional ability scores are reported in Figure 2, and the mean values were 3.3 and 1.9, respectively.

At follow-up, the mean values of the physical performance and functional ability scores were 4.6 and 3.4, respectively; Figure 3 shows the distribution of the difference between the baseline and follow-up scores.

These changes were more evident in people with poor PHS compared to people with good PHS, although the difference did not reach statistical significance, as shown in Table 2.

The incidence of loss of physical performance or of functional ability was 56.4% and 51%, respectively. The incidence of loss of physical performance was 51.7% among participants with good PHS and 59.6% among those with poor PHS (Figure 3). Participants with good PHS did not have a change in risk for a loss of physical performance (RR: 0.87, 95% CI: 0.64–1.17). Adjustment for age, gender, baseline physical performance, depression, the number of comorbidities, education, financial income, and social support, however, showed that these participants had a 34% reduction in the risk for this outcome (95% CI: 0.41–1.06) (Table 3).

Functional ability worsened in 40% of participants with good PHS and 58.4% in those with poor PHS (Figure 4), and the corresponding relative risk was 0.68 (95% CI: 0.48–0.98). When adjusting for age, gender, baseline functional ability, depression, the number of comorbidities, education, financial income, and social support, the relative risk was 0.64 (95% CI: 0.40–1.04) (Table 3).

## 4. Discussion

Our study shows that a good PHS can be used to identify older adults at a lower risk of worsening their physical performance and losing personal independence after a hip fracture.

These results are in line with prior research showing that PHS is a useful construct to identify people at a higher risk of poor outcomes, including not only hospitalizations [10] and mortality [8] but also worse functional status. In fact, Tomioka et al. reported that self-rated health can predict a decline in instrumental activities of daily living (IADLs) in a large sample of community-dwelling older persons from Japan who were independent in IADLs at baseline and were followed up for 3 years [20]. PHS has also been identified as a predictor of decline in basic activities of daily living in functional independent community-dwelling older adults [21]. However, none of these studies evaluated the effects of PHS in people with a stressful, health-related event and therefore provided no information on resilience [6].

In this context, Mennig et al. showed similar results 12 months after an elective surgery in a population of 1580 older adults: PHS can predict changes in functional health, evaluated with the Barthel index [22]. However, it is likely that people undergoing elective surgery may be more independent and may have a better health status with respect to older adults in whom occurred a hip fracture, which is an unforeseeable and major event with a dramatic impact on physical function and is more common among people with worse physical function [23]. Thus, to the best of our knowledge, this is the first study evaluating this relationship in older adults after an acute stressor like a hip fracture.

Our results are independent of the number of coexisting diseases, depression, social support, and income. As recent studies on resilience suggest that a number of pre-stressor factors, including psychological, social and environmental factors, can predict resilience [24], this finding seems to indicate that PHS may capture the whole array of contextual situations that make a person resilient, including psychological and social factors, and may potentially serve as a comprehensive measure for predicting higher resilience levels in response to future health stressors.

People with a good PHS are likely to possess traits such as optimism, self-efficacy, and adaptive coping strategies; previous research has shown that these traits can significantly impact resilience, functional outcomes, and health trajectories [2,4,25]. Thus, psychological aspects of PHS could also play a crucial role in enhancing an individual’s resilience and improving recovery outcomes.

Recent studies on resilience suggest that pre-stressor factors can predict resilience, which include psychological, social, and environmental factors [24]. In this context, PHS may constitute such a pre-stressor resilience factor enabling the prediction of individuals’ resilience levels in response to health stressors later in life.

### Limitations

We have no information on the exact timing of the hip fracture based on these interviews; this lack of precise timing information could affect the interpretation of our results: the actual impact of the hip fracture on functional outcomes may vary depending on how recently the fracture occurred before the follow-up interview and could introduce a bias in our recovery assessment. However, we have no reason to believe that PHS may impact the timing between the hip fracture and the interview; therefore, it is sensible to think that the average time since fracture is similar between people with or without a good PHS.

Furthermore, we have no information about rehabilitative interventions after the fracture. This might affect our results since rehabilitation has a well-known effect on the recovery outcomes of older adults after a hip fracture.

PHS is a self-reported measure and may therefore be related to social desirability bias; participants may overrate their perceived health status to project a more positive image of themselves. This potential bias was mitigated by ensuring anonymous data collection, reducing the pressure to present oneself in a socially desirable way. As discussed above, psychological resilience is associated with functional outcomes, such as gait speed, walking distance, and independence in the functional abilities. Hence, optimism and positive health perception are integral to our findings, rather than being limiting factors. Finally, in our sample, the functional ability score is heavily skewed towards lower values. This flooring effect is known [24,26] and may potentially impact the sensitivity of this measure to detect differences over time. In our sample, however, the distribution of the change in this score over time is normal; therefore, it is unlikely that the uneven distribution of the underlying variable has introduced a bias in our estimates.

## 5. Conclusions

This study extends the understanding of PHS and resilience by demonstrating that a simple question about PHS is associated with the resilience of older adults after a hip fracture. Thus, a systematic question on PHS can help identify patients with higher probabilities of regaining function and those at a higher risk of worse functional outcomes after a hip fracture, thus providing useful information for resource allocation.

Future research should confirm our results in a larger population of older adults, overcoming the limitations of our study, such as the lack of data on participants’ rehabilitation programs (type and duration) and the exact timing of the hip fracture.

## Figures and Tables

**Figure 1 medicina-60-01621-f001:**
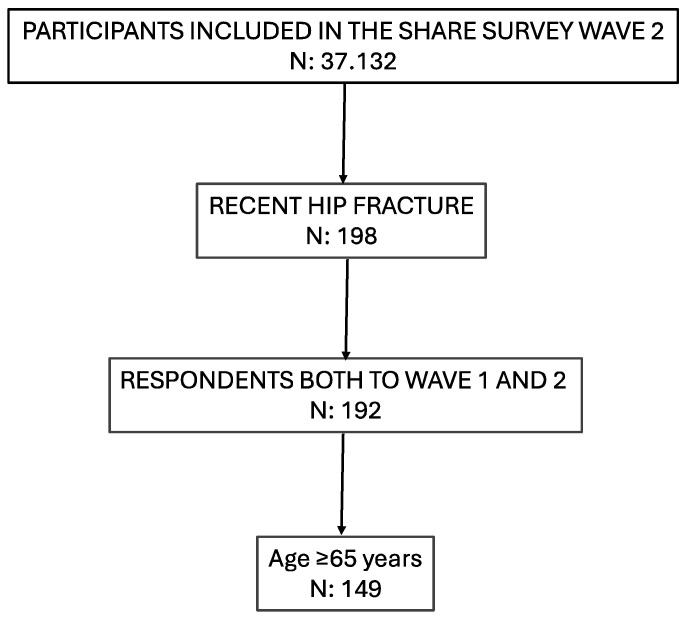
Flow chart of the study participant selection.

**Figure 2 medicina-60-01621-f002:**
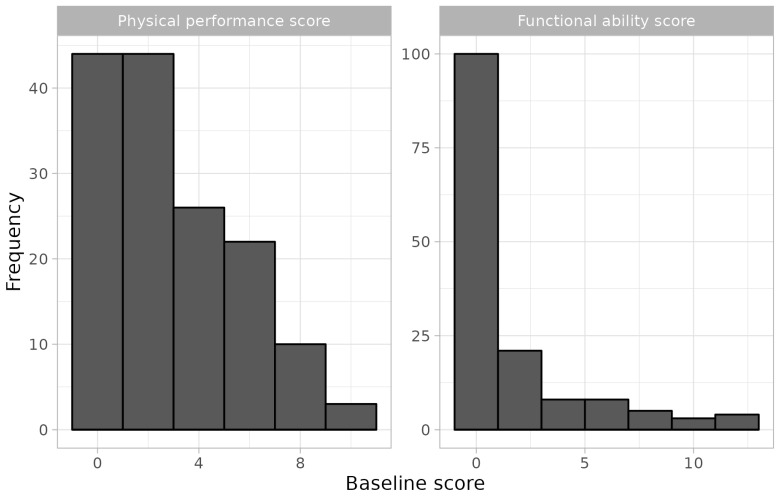
Distribution of physical performance score and functional ability score at baseline.

**Figure 3 medicina-60-01621-f003:**
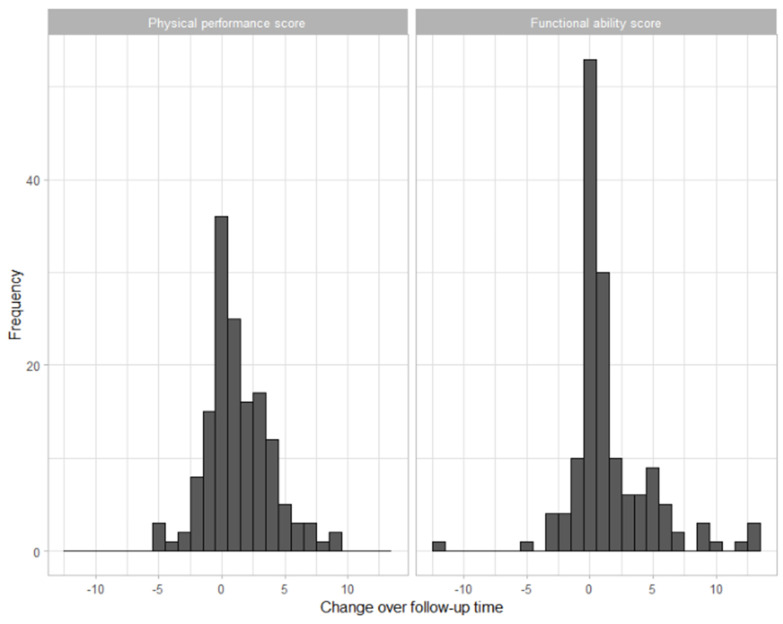
Distribution of the changes over follow-up time of the physical performance and functional ability scores.

**Figure 4 medicina-60-01621-f004:**
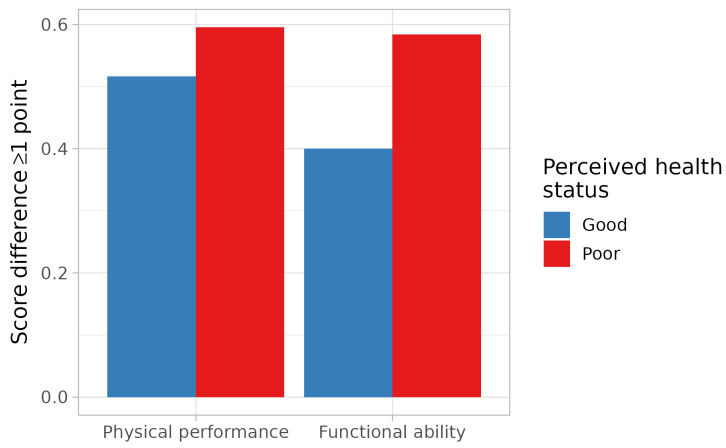
Incidence of loss of physical performance or functional ability.

**Table 1 medicina-60-01621-t001:** Characteristics of the study sample at baseline according to self-perceived health status.

	Good *n*: 60 Mean (SD) or %	Poor *n*: 89 Mean (SD) or %
Age	76.30 (7.3)	75.28 (6.0)
Sex (female)	63.3	68.5
Functional ability score	0.73 (1.8)	2.70 (3.5)
Physical function performance score	1.80 (1.8)	4.33 (2.7)
EUROD score	2.10 (1.9)	4.80 (2.6)
Number of diseases	2.18 (1.3)	3.13 (1.8)
Number of diseases pharmacologically treated	1.62 (1.2)	3.31 (2.0)
Social support		
No need	71.7	39.3
Help	20	49.4
No help	8.3	11.2
Marital status (married or partnership)	45	44.9
Smoking		
Former smoker	26.7	20.2
Smoker	18.3	13.5
Education (high school)	31.7	24.1
Net income	16,112.74 (14,513.9)	10,363.45 (7989.3)

**Table 2 medicina-60-01621-t002:** Changes in physical performance and functional ability scores according to perceived health status.

	Good PHS *n*: 60	Poor PHS *n*: 89	*p*-Value
Change in physical performance score (mean [SD])	1.20 (2.48)	1.33 (2.59)	0.398
Change in functional ability score (mean [SD])	1.32 (2.99)	1.61 (3.51)	0.217

**Table 3 medicina-60-01621-t003:** Crude and adjusted relative risk for worsening physical performance or functional ability scores.

	Crude	Adjusted
Physical performance score	0.87 (0.64–1.17)	0.66 (0.41–1.06) *
Functional ability score	0.68 (0.48–0.98)	0.64 (0.40–1.04) **

* Adjusted for age, gender, baseline physical performance, depression, number of comorbidities, education, financial income, and social support. ** Adjusted for age, gender, baseline functional ability, depression, number of comorbidities, education, financial income, and social support.

## Data Availability

Data are freely downloadable on the web at the site https://share-eric.eu.

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
