# Peer review of "Perceived Health Status Predicts Resilience after Hip Fracture in Older People"

_medicina, 2024, doi:10.3390/medicina60101621_

Round 1

Reviewer 1 Report

Comments and Suggestions for Authors

Dear authors,
First of all, I would like to congratulate you for your work. You touched on an important point. He emphasizes that if the perceived health status is better, patients feel less functional limitation even when they suffer a hip fracture. There are some shortcomings in your article. First of all, the tables are not organized enough. More data should be added to the tables so that they reflect more data. Additionally, there are spelling errors in the article and they need to be corrected and edited by a native speaker. I would like to re-review the article after these edits are made.

Comments on the Quality of English Language

It must be improved

Author Response

Dear authors,
First of all, I would like to congratulate you for your work. You touched on an important point. He emphasizes that if the perceived health status is better, patients feel less functional limitation even when they suffer a hip fracture. There are some shortcomings in your article.

First of all, the tables are not organized enough. More data should be added to the tables so that they reflect more data.

We thank the reviewer for this suggestion, We expanded table 1 including variables on number of diseases pharmacologically treated, social support, marital status, smoking.

Additionally, there are spelling errors in the article and they need to be corrected and edited by a native speaker. I would like to re-review the article after these edits are made.

We apologize for the spelling errors. The text was thoroughly reviewed for grammar and readability.

Reviewer 2 Report

Comments and Suggestions for Authors

Recommendations:

1. The rationale behind the dichotomisation of PHS should be clarified, and scales should be described consistently in different sections of the manuscript. Furthermore, the authors should discuss potential biases that may arise from self-report measures of PHS and how they are addressed.

2. More details about sample size calculation and whether statistical power is sufficient for detecting significant differences need to be provided. In particular, expand on the baseline characteristics, inclusion criteria, and exclusion criteria, as well as their possible effects on generalisability.

3. The limitation regarding the timing of hip fractures and its effect on results could have been discussed more extensively. There should also be sensitivity analyses to adjust for possible variations in fracture timing across survey waves.

4. Functional ability scores and disability scores should be explained further by providing more details on how they are calculated, as well as their clinical relevance. Moreover, there needs to be a discussion regarding any likely floor or ceiling effects in these scores that may impact study findings.

5. Other confounders such as prior functional status, social support, and rehabilitation services received, which might affect recovery outcomes, ought to be considered by researchers.

6. The implications for resilience research are deepened by analysing findings, considering broader contexts.

7. Discuss how the findings fit within the broader context of resilience research and their implications for clinical practice. Explore the potential mechanisms by which PHS influences resilience and recovery, integrating insights from psychological and social factors.

8. While some limitations are acknowledged, a more comprehensive discussion of potential biases, measurement errors, and their impact on the study conclusions is necessary. Suggest directions for future research to address these limitations and further validate the findings.

The findings highlight the importance of assessing PHS as part of routine evaluations in older adults, particularly those at risk of hip fractures. This could lead to more personalised and targeted rehabilitation strategies. The study opens avenues for future research on resilience in older adults, encouraging more comprehensive and longitudinal studies to validate and expand upon these findings.

Comments on the Quality of English Language

Minor editing is required.

Author Response

  1. The rationale behind the dichotomisation of PHS should be clarified, and scales should be described consistently in different sections of the manuscript. Furthermore, the authors should discuss potential biases that may arise from self-report measures of PHS and how they are addressed.

We thank the reviewer for the suggestion. We explained in the Methods (page 6, lines 111-114) that PHS was dichotomized because we were conceptually interested in the presence/absence of a good PHS.  

We revised the manuscript to have a consistent terminology with respect to our outcomes.

The potential bias related to self-reported PHS have been addressed in the Discussion (page 18, lines 278-283).

  1. More details about sample size calculation and whether statistical power is sufficient for detecting significant differences need to be provided. In particular, expand on the baseline characteristics, inclusion criteria, and exclusion criteria, as well as their possible effects on generalisability.

This is an exploratory study that makes use of a publicly available dataset. As there has been no active enrolment of patients, no preliminary sample size calculation has been performed, and ours can be considered a convenience sample. We acknowledge that given the relatively small sample size, the study is underpowered, and it is likely that we would have obtained narrower confidence intervals with a larger sample size. In this context, however, this is a conservative bias, and we believe that our results point nonetheless towards an association between PHS and resilience.

We provided more information on the SHARE study inclusion and exclusion criteria (page 5, lines 89-94) and expanded the description of the sample (Table 1).

  1. The limitation regarding the timing of hip fractures and its effect on results could have been discussed more extensively. There should also be sensitivity analyses to adjust for possible variations in fracture timing across survey waves.

Unfortunately, we have no information on the timing of hip fracture. We better discussed it in the limitations of the study (page 17, lines 266-272). We used data only from waves 1 and 2, and considered hip fractures occurred in this time lapse. Therefore, we could not perform a sensitivity analysis to adjust for possible variations in fracture timing across survey waves.

  1. Functional ability scores and disability scores should be explained further by providing more details on how they are calculated, as well as their clinical relevance. Moreover, there needs to be a discussion regarding any likely floor or ceiling effects in these scores that may impact study findings.

We better described these two scores in the Methods (pages 6-7, lines 117-142). Disability score was renamed as Functional ability score and Physical function was renamed as Physical performance score.

The potential impact of any floor or ceiling effects was discussed in the Discussion (page 18, lines 283-288).

  1. Other confounders such as prior functional status, social support, and rehabilitation services received, which might affect recovery outcomes, ought to be considered by researchers.

As suggested, we included social support as an additional confounder in our models. Our results are not significantly changed (table 3).

Unfortunately, we have no information about a possible rehabilitation program received after the hip fracture; we added this to the limitations of our study (page 18, lines 273-275).

We agree with the reviewer that prior functional status may affect our outcomes; the models were adjusted for the relevant baseline score (see Methods, pages 8-9, lines 172-176, and table 3).

  1. The implications for resilience research are deepened by analysing findings, considering broader contexts.

We discussed it more extensively in the Discussion (pages 16-17, lines 258-265).

  1. Discuss how the findings fit within the broader context of resilience research and their implications for clinical practice. Explore the potential mechanisms by which PHS influences resilience and recovery, integrating insights from psychological and social factors.

We discussed it more extensively in the Discussion (pages 16-17, lines 258-265) and Conclusions (page 19, lines 283-290).

  1. While some limitations are acknowledged, a more comprehensive discussion of potential biases, measurement errors, and their impact on the study conclusions is necessary. Suggest directions for future research to address these limitations and further validate the findings.

The limitations are now largely discussed in the Discussion (pages 17-18, lines 260-281). We also modified the Conclusions accordingly (page 19, lines 291-298).

The findings highlight the importance of assessing PHS as part of routine evaluations in older adults, particularly those at risk of hip fractures. This could lead to more personalised and targeted rehabilitation strategies. The study opens avenues for future research on resilience in older adults, encouraging more comprehensive and longitudinal studies to validate and expand upon these findings.

Round 2

Reviewer 1 Report

Comments and Suggestions for Authors

Dear authors,

You have made the revisions requested by me. There is no need to further review. Congratulations.

Best regards.

Author Response

Dear Editor,
We thank you for the comments and for the opportunity of further improving our paper.  We modified the manuscript according to your comments and we hope that it is now suitable for publication.

Please find below the point-by-point response to the comments.

Regards,

Claudio Pedone

First, it seems that the authors used and analyzed very old data coming from 2004-2006 and 2006-2007. As now we have year 2024, how so old data could reflect the current situation? During a span of almost 20 years, a lot could change. Would it be possible to include more recent data.

We had to resort to the first two waves of the SHARE dataset because they were those providing the greatest number of fractures. While it is true that several things may change over such a long time frame, we believe that it is unlikely that the relationship under study is subject to significant secular trends. As we reported in the Discussion, a good PHS reflect a positive attitude that has been shown to positively impact health outcomes. This relationship between mind set and health has been anecdotally since time out of mind and has been reported in the scientific literature from different decades (J Gerontol B Psychol Sci Soc Sci 1995; 50: S344-53, Am J Epidemiol 2000; 152: 874-83, Age Ageing. 2017; 46: 265-270, Health Promotion Perspectives 2022; 12: 37-44). We added a limitation to the Discussion to address this issue.

Second, the authors got insignificant RRs. The only one significant RR was a crude RR of 0.68 (0.48 - 0.98) for functional ability. Therefore, this makes the conclusions incorrect, speculative and misleading. These two serious drawbacks have to be explained and corrected (especially that one one regarding interpretation of RRs) by the authors before taking a decision for this manuscript.

The interpretation of confidence intervals (or P-values, for that matter) as “significant” or “not significant” has been criticized for decades as being incorrect and misleading (among many, see the statement from the American Statistical Association and supporting material at https://amstat.tandfonline.com/doi/full/10.1080/00031305.2016.1154108 or Rothman K, “A show of confidence”, NEJM 1978). Confidence intervals express the precision of the estimate and provide a range of “credible” values compatible with the data. Our point estimates indicate a clinically important reduction of the risk, and the “non-significant” confidence interval show that the range of possibility consistent with the data generally indicate the presence of an association. We revised the wording of the discussion to address the reviewer’s criticism.